# Uptake of reproductive, maternal and child health services during the first year of the COVID-19 pandemic in Uganda: A mixed methods study

**Simon P. S. Kibira**[1]☯*, **Emily Evens**[2]☯, **Lilian Giibwa**[3]☯, **Doreen Tuhebwe**[4],
**Andres Martinez**[5], **Rogers Kagimu**[6], **Charles Olaro**[7], **Frederick Mubiru**[8],
**Samantha Archie**[2], **Rawlance Ndejjo**[9], **Noel Namuhani**[4], **Martha Akulume**[4],
**Sarah Nabukeera**[3], **Rhoda K. Wanyenze**[9], **Fredrick E. Makumbi**[3]☯

1 Department of Community Health and Behavioural Science, School of Public Health, College of Health Sciences, Makerere University, Kampala, Uganda, 2 Health Services Research Division, FHI 360., Durham, NC, United States of America, 3 Department of Epidemiology and Biostatistics, School of Public Health, College of Health Sciences, Makerere University, Kampala, Uganda, 4 Department of Health Policy Planning and Management, School of Public Health, Makerere University, Kampala, Uganda, 5 Behavioral, Epidemiological and Clinical Sciences Division, FHI 360., Durham, NC, United States of America, 6 Department of Planning, Financing and Policy, Division of Health Information Management, Ministry of Health, Kampala, Uganda, 7 Directorate of Clinical Services, Curative, Ministry of Health, Kampala, Uganda, 8 Research Utilization Department, FHI 360., Durham, NC, United States of America, 9 Department of Disease Control and Environmental Health, School of Public Health, College of Health Sciences, Makerere University, Kampala, Uganda

☯ These authors contributed equally to this work.
* pskibira@musph.ac.ug

**Data Availability Statement:** The dataset used for this paper is from the National Health Information System. This can be accessed on request to the

## Abstract

Use of reproductive health (RH), maternal, newborn and child health (MNCH) services in Uganda is suboptimal. Reasons for this are complex; however, service-delivery factors such as availability, quality, staffing, and supplies, contribute substantially to low uptake. The COVID-19 pandemic threatened to exacerbate existing challenges to delivery and use of high-quality RH and MNCH services. We conducted a mixed methods study, combining secondary analysis of routine electronic health management information system (eHMIS) data with exploratory key informant interviews (KII) to examine changes in health service uptake over the course of the pandemic and to understand service delivery adaptations implemented in response. We analyzed eHMIS data for four services (family planning, facility-based deliveries, antenatal visits, and immunization for children by one year), comparing them across four time periods: pre-COVID-19, partial lockdown, total lockdown and post lockdown. Additionally, KIIs were used to document adaptations made for continuity of health services. Use of services declined substantially during total lockdown; however, rebounded quickly to earlier observed levels, during the post lockdown for all four services, especially for immunization for children by one year. KIIs identified several health services delivery adaptations. At the community level, these included: community outreaches, training some mothers as community liaisons to encourage others to seek health services, and support from local leaders to create call centers to facilitate clients transport during travel

permanent secretary, through the Director, Curative Services at the Ministry of Health, and through the Assistant Commissioners: Division of Health Information Management, email: paul. mbaka@health.go.ug, and the Division of Reproductive and Infant Health email: richard. mugahi@health.go.ug.

**Funding:** This research was funded under the Grant funded by Family Health International (FHI) under Cooperative Agreement/Grant no 7200AA19CA00041 funded by United States Agency for International Development (USAID). The content of this paper does not necessarily reflect the views, analysis of policies of FHI 360 or USAID, nor does any mention of trade names, commercial products, or organizations imply endorsement by FHI 360 or USAID. RKW. The funders had no role in study design, data collection and analysis, decision to publish, or preparation of the manuscript.

**Competing interests:** The authors have declared that no competing interests exist.

restrictions. Health facilities creatively used space to accommodate social distancing and shifted providers' roles. District leadership reassigned health workers to facilities closest to their homes, provided vehicle passes to staff, and ambulances to transport pregnant women in critical need. WhatsApp groups facilitated communication at district level and enabled redistribution of supplies. Ministry of Health produced critical guidelines for continuity of health services. Implementing partners provided and redistributed commodities and personal protective equipment, and provided technical support, training and transport.

## Introduction

Uganda, like many low-income countries, experience challenges in providing health services, including family planning (FP) and maternal, newborn and child health (MNCH) services. Karamagi et al. described the demand for essential services, access to essential services, quality of care and resilience of the health system as the four capacities of a functional health system [1]. In many countries of the World Health Organization Afro region, accessing essential services is reported as the rate limiting capacity for the system functionality [1]. Given pre-existing access challenges, there were questions of whether the COVID-19 pandemic exacerbated the already weak health system in the region, including Uganda, threatening universal access goals.

Uganda reported its first case of COVID-19 on March 21, 2020 [2] and the first death on July 15, 2020 [3]. The country took stringent measures to combat COVID-19 including being among the first African countries to impose travel restrictions, curfews, ban large public gatherings and close schools [2]. These lockdown measures were enacted on March 18, 2020, even before a case was reported in the country, and were strictly enforced [2, 3]. By June 25th 2022, Uganda had reported over 165,607 cases and 3,613 confirmed deaths [4]. The measures instituted likely helped curb the pandemic, but could also have had negative effects [5] including: limiting clients' access to health services, providers' ability to provide care, constricting supply chain [6], and posing lasting damage to livelihoods and education [7, 8]. Additionally, fear of COVID infection, lack of personal protective equipment, restrictions on gathering, and space challenges in health facilities and commodity storehouses placed further stress on the health system.

Previous research has documented negative effects of epidemics including Ebola Virus Disease [9] and COVID-19 [10–12] on health systems as well as the delivery of sexual and reproductive and maternal and child health services [10, 13]. Ensuring continuity of non-COVID-19 health services, including FP and MNCH services is essential especially in low-income settings where need for these services is significant and system challenges are more common. When disrupted over an extended period, limitations in access to FP care in resource limited settings can impact maternal and infant mortality [9].

Analyzing the effect of COVID-19 on the utilization of FP and MNCH services in Uganda and documenting the adaptations implemented to minimize disruptions in service delivery is essential to learn from the COVID-19 pandemic experience and prepare for future shocks to the health care system in this setting [11]. Furthermore, there has been a call to action to document these adaptations for continued learning [14], and some work on this is already happening in Uganda [15].

The goal of this study was to track the impact of COVID-19 on the utilization of four FP/MNCH services at national level: family planning service visits, facility-based deliveries,

attendance of at least four antenatal visits, and full child immunization by one year of age. We identified four key time periods: pre-COVID-19, partial lockdown, total lockdown and post lockdown. Additionally, the study sought to document adaptations made by the Government of Uganda and partners at the community, facility, district, and national levels to ensure continuity of health services during COVID-19 through qualitative interviews. The lessons obtained from this study are important in building a resilient system to absorb future shocks.

## Materials and methods

### Ethics statement

The study received human subjects' ethics approval from the FHI 360 Protection of Human Subjects Committee, the Makerere University School of Public Health Research and Ethics Committee (Protocol 884) and the Uganda National Council for Science and Technology (HS1031ES), as well as permission to use the National electronic Health Management Information System (eHMIS) data from the Uganda Ministry of Health. All key informants in the study provided consent to an audio recorded telephone interview.

This was a mixed methods study that included a secondary analysis of routine FP, and MNCH service delivery data and qualitative key informant interviews (KIIs). The secondary analysis focused on women of reproductive age (15 to 49 years) and children aged one year. Uganda operates local governments that are called districts. These are subdivided into sub counties, parishes and villages. Within the districts, the public health facilities serve a health sub district which can include more than one sub county (Health center IV), sub county (health center III), parish (health center II and village (Village health teams). The district health officer is one of the technical heads at the district, overseeing the health services. The public sector includes the village health team that function as level I. These report to the health centers, and they offer minimal FP services and referrals for mothers as well as supporting mobilization services. The static public facilities are health center II, III, IV and hospitals [16]. The private include clinics and hospitals. The health services reported in this paper are offered at all levels with different levels of complexity. All the health centers and hospitals offer ANC, delivery services, immunization and family planning services, but with varied levels of complexity.

### Quantitative methods

We used routine data abstracted from the eHMIS. The data included the monthly total number of FP and MNCH service visits starting from January 2018 to December 2020, for all facilities reporting to the eHMIS. The data abstracted were reviewed to identify and address inconsistencies. We identified outliers using a scatter plot of the monthly total number of services within each facility. Outliers were replaced with the average of two observations from the two consecutive months prior to the outlier and the two most immediate after the outlier months.

All public and private facilities in the country providing MNCH services, that report in the electronic eHMIS were considered. These included clinics, health centers and hospitals. Monthly reports for this paper are from 7,780 facilities.

Analysis used four key indicators: total number of FP service visits per month, total number of a fourth or higher antenatal care visit per month, total number of women delivering in facilities per month, and total number of children fully immunized by one year of age, per month. For this analysis, the monthly total number of health services for each of the examined services was aggregated across all reporting health facilities to form a monthly national level count for each of the 36 months between 2018–2020.

To examine changes in service use associated with the COVID-19 restrictions over the course of the 36 study months, we categorized the time into four periods: 1) pre-COVID-19, from January 2018 through December 2019; 2) partial lockdown, from January 2020 through March 2020; 3) total lockdown, during April 2020; and 4) post-lockdown, from May 2020 through December 2020. The pre-COVID-19 period was a time prior to the declaration of COVID-19 as a pandemic. The partial lockdown period was a time during which COVID-19 awareness was increasing in Uganda, World Health Organization declared the pandemic, Uganda's first case was announced, and partial restrictions including closing schools and public transport were declared. The period of total lockdown included total ban on all modes of transport except essential services, institution of a dusk to dawn curfew, and no gatherings were allowed. The post-lockdown included easing of travel restrictions allowing limited use of private cars, and provision of guidelines on the continuity of essential health services from the MOH as well as the continuation of a curfew.

We conducted a descriptive trend analysis for each key outcome, at the national level, over the 36 months then fitted a separate regression model to each outcome to estimate the effect of the lockdown on the utilization of services. The resulting descriptive graphical analysis displayed trend and seasonal variation; therefore, the regression models were adjusted for pre-existing trends and seasonality effects. To determine the nature of trends over time, a locally weighted scatterplot-smoothing (LOWESS) curve was plotted for each outcome to establish changes in slopes throughout the 36-month study period. In case of change in slope, we visually identified the points at which the trend changed (knots) and created piecewise trends that were used in the regression model. For the LOWESS curves that portrayed a quadratic nature, we added a quadratic term to the model and compared the resulting model with the model without the quadratic term using the Akaike Information Criteria (AIC) to aid model selection. The preliminary descriptive analysis also showed that the variance of each key outcome was significantly larger than the mean, suggesting over-dispersion. These being count data, we opted for a negative binomial regression model over a Poisson regression model, so as to address the over-dispersion and included Huber-White sandwich estimators (robust standard errors) to cater for violation of any underlying assumptions. To capture seasonality, the model included a dummy for calendar month coded "0" for January as a reference for the other 11 months February-December. Each model provides estimates of the monthly total number of services during each of the three periods (partial lockdown, the total lockdown, and the post-lockdown periods) relative to the pre-COVID-19 period, adjusting for seasonality, and trends. These results are presented as incidence rate ratios (IRRs) with their corresponding 95% confidence intervals. All analyses were done using STATA version 14.2.

## Qualitative methods

We conducted KIIs with a sample of individuals who were purposefully selected from eight districts in northern, eastern, central, and western regions, stratified by urban/rural setting. Using the 2019 district league table ranking [17], a performance score assigned by the Uganda Ministry of Health before COVID-19, based on utilization of services and health system resources available in each district, we selected four of the top districts (Serere and Jinja from eastern, Bushenyi from western, and Masaka from central) and four of the bottom districts (Nabilatuk, and Amudat from northern, Mubende from central and Kakumiro from western).

Participants for KIIs in each district included six groups: 1) health facility in-charges from the largest health center or hospital; 2) focal persons in charge of reproductive and child health; 3) commodity storehouse managers; 4) community health workers, locally known as village health team (VHT) members, who specialize in MNCH and FP and support the catchment

areas of the facility where the interviewed health facility in-charge works, 5) representatives from a major private providers; and 6) representatives from the major implementing partners (IPs), as identified by the district health officers. District and facility in-charge participants were identified in collaboration with District Health Officers who provided names of potential participants believed to have expertise and knowledge of reproductive health, MNCH, and FP services, while VHTs were identified by facility in-charges. All KII participants had worked in their position for at least three years and were actively providing service during the March and April 2020 COVID-19 lockdown period.

A pretest of the interview guides was conducted by the research team in three non-study districts; Kampala city, Kamuli, and Jinja. All pretest interviews were conducted via phone, and audio recorded with consent, to assess feasibility of audio recording using the phone. After each interview, a template was completed to inform group learning around the scheduling of the interviews, the consenting process, duration, clarity of questions and any explicit respondent feedback on the interview experience. There were no major revisions to the content after the pretest, apart from reordering questions to improve the flow, defining the terms "pre-COVID, COVID and post- COVID" periods, and operationalizing the word "adaptations" especially for the VHTs.

Experienced interviewers trained on research ethics, study procedures and use of interview guides conducted interviews. Interviews were conducted in the December 2020. All interviews were conducted in English or a local language (for some VHTs) using a translated tool, via phone by two researchers—an interviewer and a note taker and were audio recorded with the participant consent. For the privacy of key informants, interviewers used ear buds and sat in a private place during the process to ensure no other person listened in. The key informants confirmed before the interview that they were in a private place and comfortable to talk. The pretested KII guide was used to facilitate the interviews and included questions on the availability of services, commodities, and human resources as well as adaptations to address the impact of COVID-19.

Audio recordings were transcribed by the notetaker. A third researcher reviewed the transcripts along with the recordings for quality assurance. Transcripts were then used to conduct an excel-based, thematic matrix analysis. The matrix included deductive themes of inquiry including challenges of service delivery during COVID-19, adaptations to continue service delivery as well as emerging themes such as COVID impact on demand for and supply of services, and any lessons from the pandemic to be applied in similar situations in the future. For approximately one-quarter of interviews, one researcher summarized the data into the matrix and a second reviewed the summaries, comparing them to the transcript to ensure inter-coder reliability. The verification by another analyst continued until all members of the qualitative research team (including SPK, EE, SN, RN, NN and DT) were satisfied that data were being coded in a similar fashion. Data on each code were then aggregated by participant category to document the experiences of each participant group. Relevant quotes were included in the matrix to support the key findings; a few typical quotes are part of the study results section of this paper.

## Results

### National trends in monthly total number of services

**Family planning service visits.**    Fig 1, shows an observed rising trend in the monthly total number of FP service visits over the study period. However, a decline in the number of FP service visits was observed in April 2020, the month of total lockdown. During the post lockdown period, the service visits rebounded to the continued trajectory witnessed in the periods before, and even surpassed the pre COVID period levels.

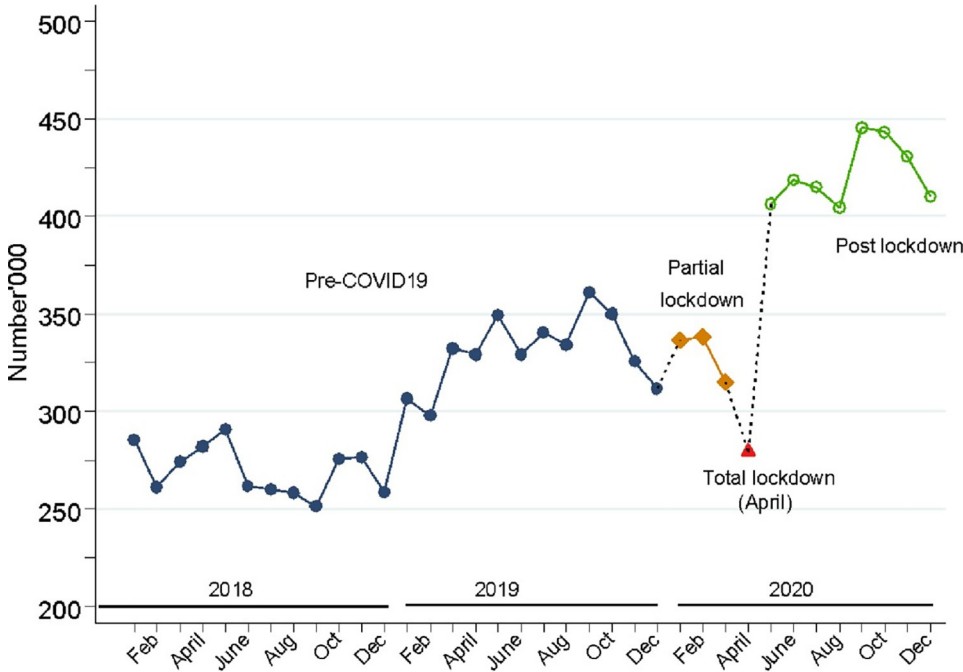

**Fig 1. Trends in monthly total number of FP service visits in Uganda, from 2018 to 2020.**

**Antenatal care visits.** Results in Fig 2, show that the monthly total number of ANC4+ visits generally increased throughout the pre-COVID19 period, although there were seasonal drops in the months of December in 2018 and 2019. During the partial lockdown and the total

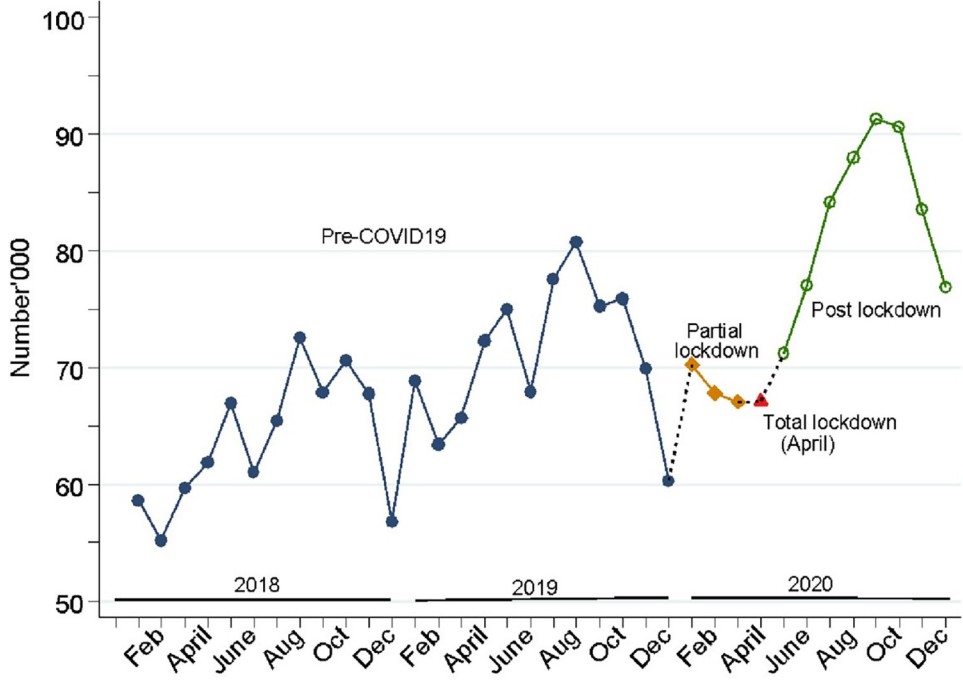

**Fig 2. Trends in monthly total number of ANC4+ visits in Uganda, from 2018 to 2020.**

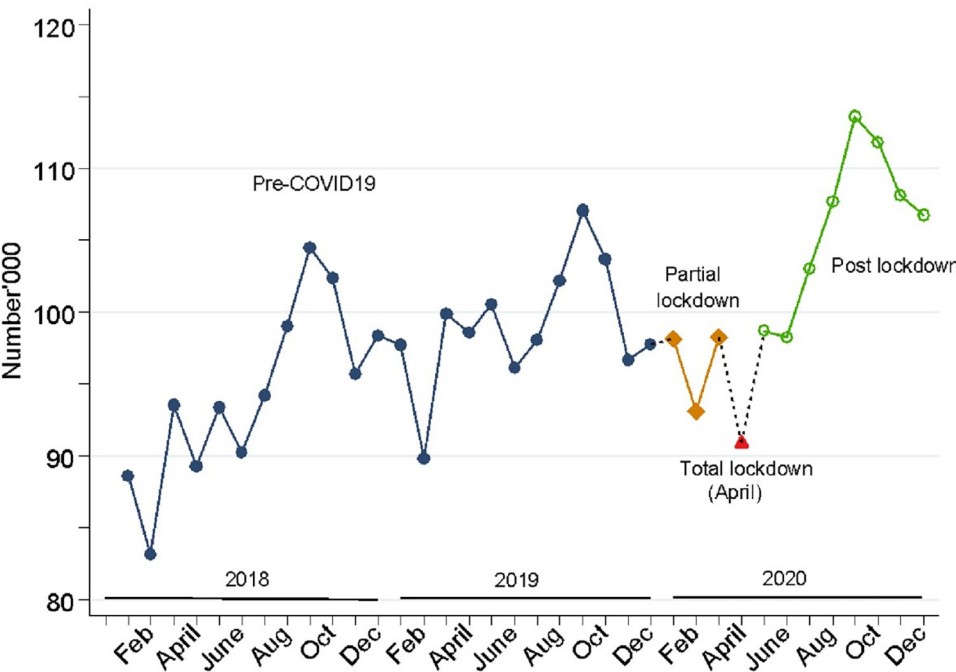

**Fig 3. Trends in monthly total number of women delivering in health facilities in Uganda, from 2018 to 2020.**

lockdown periods, there were declines not observed in the years before. However, the monthly total number of ANC4+ visits also continued to rise thereafter, following a similar trend as in the pre-COVID19 period.

**Women delivering in health facilities.** Fig 3, shows the monthly total number of women delivering in health facilities increased gradually over time during the pre-COVID19 period. The number declined during the partial lockdown period, mostly in February, a month which experienced similar drops in the previous years. In 2020, the total number of women delivering in facilities was lowest in April, the month with total lockdown. However, they increased steadily thereafter peaking in the month of September before declining again, although never to levels in the pre COVID period.

**Children fully immunized by 1 year of age.** Results in Fig 4, show that the trend in monthly total number of children fully immunized by 1 year of age showed an increase gradually over time during the pre-COVID19 period. The total number steadily declined for the first three months of 2020 in the partial lockdown period, and with the lowest dip observed in April 2020, the month of total lockdown. The results indicate that the numbers fully recovered during the post lock down period to follow the trends observed before COVID, rising much higher than any other period of the study until November 2020. However, in December 2020 there was a seasonal drop observed in corresponding years.

**Changes in the four outcomes in 2020 relative to the pre-COVID-19 period.** Table 1, shows that relative to the pre-COVID period, two outcomes, monthly total number of FP service visits and ANC4+ visits declined significantly by 17% and 6% respectively during the partial lockdown period. However, the other two, facility-based deliveries and full child immunization by one year did not differ from the reference period, during the same period. During the lockdown period, relative to the pre-COVID19 period, only child full immunization did not differ significantly while the other three outcomes (FP service visits, ANC4+ and facility-based deliveries) declined significantly by 34%, 14% and 8%, respectively. Only child

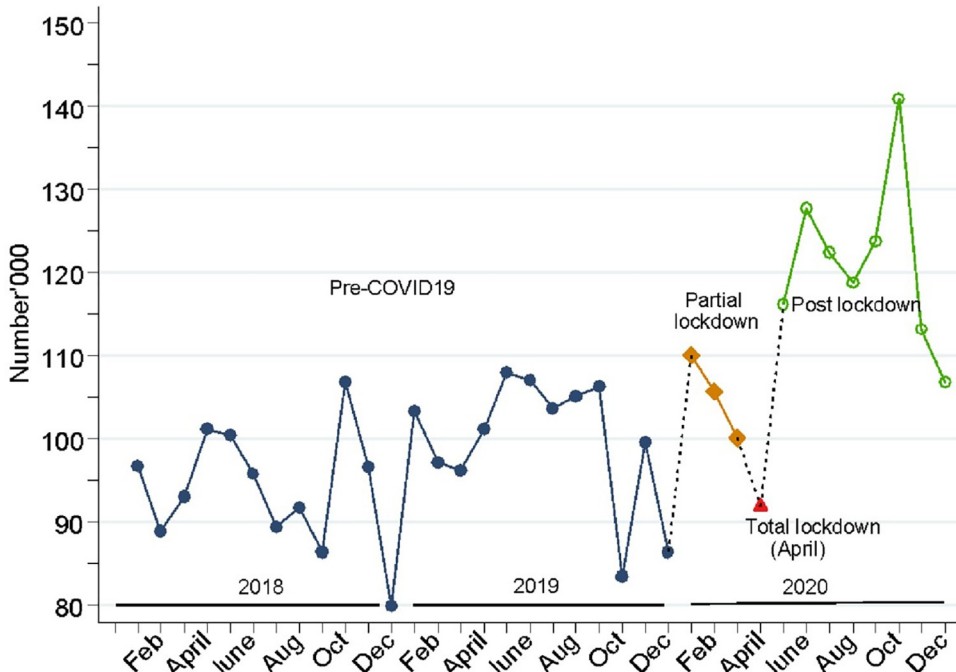

**Fig 4. Trends in monthly total number of children fully immunized by 1 year in Uganda, from 2018 to 2020.**

full immunization increased significantly during the post lockdown period while the other three outcomes did not significantly differ, relative to the pre-COVID19 period.

## Adaptations to ensure continuity of health services during COVID-19

A total of 50 KIIs were conducted, including at least one interview per participant type across the eight districts. Eight interviews were from health facility in-charges, focal persons, commodity storehouse managers, community health workers, and private providers, and 10 from implementing partners. The KIIs explored innovations and adaptations made at the community, facility, district, and national levels to ensure continuity of health services during COVID-19.

Community outreaches were seen as the most successful adaptations across all participant categories. While outreaches existed prior to the pandemic, participants noted that outreaches were initially halted in the early stages of the pandemic due to gathering restrictions. When guidance from the Ministry of Health later allowed implementing partners to resume

**Table 1. Incidence rate ratios and percent change in key outcomes during 2020 study period, relative to the pre-COVID-19 period (January 2018—December 2019).**

|  | FP service visits | ANC-4 visits | Facility-based deliveries | Full immunization for children by one year |
|---|---|---|---|---|
| **Reference:** Pre COVID19 | **IRR (95%CI)** | **IRR (95%CI)** | **IRR (95%CI)** | **IRR (95%CI)** |
| Partial lockdown | 0.83 (0.77,0.99) | 0.94 (0.91,0.98) | 0.99 (0.95, 1.02) | 1.10 (0.96, 1.26) |
| Total lockdown | 0.66 (0.63,0.69) | 0.86 (0.83,0.89) | 0.92 (0.87, 0.96) | 0.94 (0.79, 1.12) |
| Post-lockdown | 0.95 (0.89,1.01) | 0.95 (0.86,1.04) | 1.03 (0.97, 1.10) | 1.26 (1.03, 1.54) |

*Significant at 5% level

The IRRs accounted for seasonality and trends

outreaches, their content was adapted to ensure the health services reached community members outside of facility settings. During outreaches, VHTs mobilized and sensitized the community about COVID-19, provided information on the continued availability of services in facilities, and encouraged people to access health care in facilities as needed. Participants described how this improved awareness in the community about the continuity of services and increased the number of clients seeking services at facilities.

> *"At first, the community members would fear to come to the outreaches, but later after sensitization with the local leaders, they understood and accepted to come for the services in the community. Even delivery of services by the VHTs was more acceptable."*—Focal Person

> *"We worked with VHTs to help mothers access facilities and also ensure there is social distance. We are also having an intervention through parish coordinators who locate all pregnant mothers in the parish."*—Implementing Partner

Other community-level interventions included: training mothers as community liaisons to encourage others to seek services and support from local leaders to create a call center to facilitate the transport of clients to facilities during travel restrictions.

Health facilities also reported adaptations. They reported creatively using existing physical space to accommodate social distancing within facilities including using tents to avoid overcrowding in the outpatient department. Facility in-charges and private providers also reported changes in roles at health facility such as: health care workers working overtime and being able to support COVID-19 response and management, nurses trained to monitor women in labor, and midwives dispensing drugs. Focal persons also noted that midwives and nurses filled in for clinical officers and in-charges.

At the district level, a key adaptation was the redistribution of commodities among public health facilities to bridge the supply gap. This was reported across all categories of participants.

> *"It is because we redistributed the excess supplies got from one facility to another that needed them and in that the gap was covered."*—Commodity Store Manager

Another district-level adaptation was the provision of extra airtime to focal persons to ensure they communicated with health facilities to encourage health care workers to continue FP service provision. Additionally, the use of mobile technology like 'WhatsApp' groups enabled health care providers to communicate remotely and ensure teams could update each other and monitor commodity stocks to facilitate redistribution. Some districts also installed information communication and technology equipment to support virtual meetings and mentorship of health care works on the continuity of services.

There were additional district-level adaptations regarding transport; these included reassigning health workers to work at facilities closest to their homes to minimize transport challenges; supporting health care workers with vehicle passes to freely move and access facilities for service provision, and provision of ambulances to transport pregnant women in critical need of care during the lockdown.

At the national level, the most commonly noted support participants received from the Ministry of Health was the provision of guidelines for continuity of essential health services [18]. These guidelines included COVID-19 prevention measures, management of COVID-19, and occupational health and safety guidance. Most participants noted that guidelines encouraged continuity of services by helping health workers learn how to protect themselves while providing services and understanding which services they could continue during the epidemic.

A few respondents mentioned that the guidelines boosted healthcare worker confidence to continue offering services and enhanced quality improvement. The guidelines were obtained from district health offices, downloaded, or received during training. Most participants reported following these guidelines.

> *"The guidelines kind of gave us a firm ground to support the health workers to continue providing the services without necessarily trying to compromise with their own health."*—Implementing partner

> *"They have helped us in such a way that in areas where we had a gap, we have been able to improve due to guidelines. So, before the guidelines, we were afraid but when the guidelines came up the services continued smoothly. The health workers knew how to protect themselves like get masks on, sanitize, so, after touching a mother they would sanitize."*—Focal person

In addition to program adaptations, implementing partners also reacted by reorienting resources and activities to support national program response to ensure continuity of services. Partner support to districts included: redistributing commodities and supplies to bridge the supply gap; provision of FP commodities, especially long-acting methods, and technical support for intrauterine device and implant insertion. Partners also supplied personal protective equipment (PPE) and drugs for other diseases, provided transport in form of motorcycles and fuel for the district health workers, and resources to train and mentor health workers through support supervision to offer services. A few focal persons also noted that IPs supported the conducting of outreaches.

> *". . .I used to liaise with those implementing partners. . . that are helping us. They used to come here and ask, 'we are going to this facility; do you have anything [to deliver]?' Actually, they are more effective in those issues of redistribution. So, you could give them commodities and they delivered to health facilities that were in the same direction."*- District commodity store manager

Qualitative interviewers also asked about key lessons learned and what needs to be done to address challenges and prepare for future challenges to the health system, including pandemics. Maintaining and strengthening community engagement through mobilization, sensitization and increasing outreaches including supporting VHTs to provide house-to-house services was noted. Ensuring health care workers are informed, equipped, and supported financially and emotionally to ensure continued service provision and prevent burnout. Ways to support health care workers included: hiring, training, and supporting additional staff, including VHTs and volunteers during the pandemic; providing training on COVID-19 standard operating procedures and how to provide mental health support for clients. Also noted were improving transport for clients and providers through ensuring availability of ambulances for emergencies, availability of other vehicles including boda bodas, providing vouchers for transport and fuel for emergencies, which would also strengthen the health system beyond pandemic times. Ensuring collaboration with private providers for service provision, commodities, and supervision was noted. Finally, ensuring commodities, PPE and other supplies are available through adequate transport for last-mile delivery, redistribution, and supporting districts to proactively address challenges through better use of service data to make projections, timely requisitions, ensuring budgets included expenses for unplanned situations, buffer stocks at the district and regional levels, bulk ordering, and increased distribution points within the community.

## Discussion

This study used routine facility level data from the Ministry of Health collected through the eHMIS to examine changes in key FP and MNCH indicators at the national level before and during the first year of the COVID-19 pandemic. Additionally, we documented service delivery adaptations from multiple perspectives to identify adaptations that supported continuity of health care service provision during this challenging time.

Our results show that for all four outcomes, national level monthly total numbers generally increased from 2018 to 2020 demonstrating the Government of Uganda and its partners' success in increasing uptake of key FP and MNCH services. This is noteworthy as accessing essential services is often reported as the rate limiting factor for a functional health care system [1]. The exceptions to this are first four months of 2020 as COVID-19 became a global challenge to the delivery and accessibility of health care and our results show declines relative to the pre-COVID-19 period. During the partial lockdown (January–March 2020), FP and ANC visits were the most affected, suffering significant declines, while there was little change in facility-based deliveries and a small, but statistically insignificant increase in full immunizations. During the total lockdown (April 2020), all four services declined, though the decline in immunizations was not significant. During post-lockdown FP, ANC and deliveries were able to rebound somewhat to levels not significantly different from pre-COVID, while immunizations had a large increase.

We hypothesize that FP and ANC-4 were the most affected services in both the partial and total lockdown periods because they may not have been considered essential by the clients seeking services. Additionally, fear of going to facilities due to perceived COVID exposure may have reduced the numbers seeking services. Deliveries may have been spared more severe impact during partial lockdown as they are a more urgent and essential service that cannot be delayed. During total lockdown facility-based deliveries could have been affected by transport restrictions before adaptations like passes for pregnant women and ambulatory services were instituted and the public made aware about them. The curfews and transport restrictions imposed as part of lockdown during March and April 2020 likely had a strong impact on uptake of all services as well as providers' ability to travel to facilities to provide care.

We identified a different pattern among the numbers of full immunization by one year which stands out for its large, significant increase during the post-lockdown period. We hypothesize that this could be due to the more flexible timing of immunizations, some can be provided anytime during a specific month or first point of contact [19], as opposed to deliveries that must happen on a specific, and uncontrolled day. Those families who postponed vaccinations in the lockdown period may have returned in higher numbers to catch up on missed vaccines for their children, driving up numbers. It also seems likely that ensuring children were fully immunized by one year was considered a priority for parents and caregivers. While both immunizations and family planning were offered during community-based outreaches, immunizations could have been viewed as more acceptable to receive within the outreach context than family planning services. Results also indicate that deliveries in October 2019 were higher than other months, and indeed children fully vaccinated at 1 year in October 2020 were at an all-time high, as well, continuing with the expected trends.

Despite the observed rising trends over the period over the three years we identified seasonal drops occurred in December for the number of FP services visits, ANC-4 visits, and children fully immunized by one year; and seasonal drops in February for the number of women delivering in health facilities. December of each year is a festive season, and this could have a seasonal impact on service uptake among services that can be postponed by a few weeks or months. There is also movement from urban to rural areas at this time, and this may impact

such services, as people wait to receive them when they return to the facilities that regularly serve them.

Understanding improvements in the numbers of all four services during post-lockdown, is important. One factor could have been the availability of guidelines on continuity of essential services that helped providers know what services were safe to provide and how to do so safely starting in the post-lockdown period. The strategies for continuity of essential health services (CEHS) were published in the month of April 2020 and implemented starting May 2020 as the total lockdown period ended and the transition to post-lockdown began [18]. Key informants emphasized that the CEHS guidelines facilitated decision making on how to ensure safe service provision. These guidelines were essential in filling the knowledge gaps for health workers and reduce the service disruptions. Recent research has found that absence of guidelines in such situations can cause panic and increase stress levels of health workers [20] ultimately affecting service provision.

Additionally, the easing of travel restrictions undoubtedly facilitated client and provider access to facilities resulting in the subsequent upturn in the monthly number of services provided during the post-lockdown period. During the total lockdown health facilities were not fully functional and both client and provider access to facilities was severely limited likely contributing to the notable drops in number of services during this period. While it is not possible to disentangle the relative impact of the provision of guidelines and the lifting of travel restrictions with our data given, they happened concurrently, assessing the benefit of travel restrictions relative to the burden they imposed for those seeking to provide and receive health care is needed to prepare rational and functional plans for future pandemics.

While we cannot identify the specific impact of adaptations on the post-lockdown increase in services, our findings suggest that adaptations implemented at the district, facility, and community levels helped minimize the impact of the COVID-19 pandemic. Qualitative findings suggest flexibility, especially at the district level, was key to ensuring communication and services continued, supplies were redistributed, health care workers were able to reach their places of work and transport for patients was available. Adaptations implemented at the community level were widely seen as having the greatest impact on services provision according to key informants. Community-based services allowed for the provision of key services while reducing crowding in facilities and ensuring patients could receive care safely and acceptability, as facilities were seen as the source of COVID-19. Additionally, community outreach educated people on COVID-19 and provided guidance on the importance of seeking facility-based care when needed. Our findings point to the importance of building systems and strengthening leadership at all levels to allow rapid adaptations and foster independence.

Our findings also point to a reliance on implementing partners who played a key role in addressing financial, transport and commodity challenges. While implementing partners are a big part of the health system in Uganda, reliance on them may not build resilient health systems in the long-term given they operate in a project mode that is timebound and often focused on vertical programs. Further, emergency adaptations for the health systems that are heavily supported by external funding as a stop gap measure may not stand the test of time when that support wanes and may not withstand future pandemics or epidemics [11]. This may leave the health system vulnerable in case of total withdrawal of implementing partners.

The study has some important limitations. The eHMIS data may be prone to accuracy challenges during capture. We minimized this by ensuring that the data were cleaned for outliers during the analysis. Due to challenges related to identifying and interviewing clients during COVID-19, we were unable to interview clients and our results lack their unique perspective on challenges or adaptations regarding access to care during lockdowns. Finally, as interviews were conducted in December 2020 regarding challenges and adaptations from the lockdown

period in March and April 2020, participants may have had difficulty in remembering events occurring during the lockdown period.

## Conclusion

Understanding what FP and MNCH services were most affected during the first year of COVID-19, along with the adaptations that supported service provision during this extremely challenging time can help build flexibility and resilience into the health system and prepare it for future shocks. Our data show that overall Uganda is on a positive trajectory of improving the number of clients who receive FP and MNCH services. Community and district level adaptations were reported as having the most impact on continuity of services while the rapid production of national guidance on continuity of health services provision reassured the health system of its priorities. Ensuring health systems can adapt locally is essential for flexibility during shocks. Additionally, improving core health system elements such as a continued focus on community engagement, ensuring health care workers are informed, equipped, and supported, improving transportation, and ensuring commodities are available is essential for a resilient system in Uganda.

## Acknowledgments

We would like to acknowledge the Ministry of Health for providing permission to access national data from the electronic health management information system. We also thank members of the technical working groups of; Family planning /reproductive health commodities security, and of the Continuity of essential health services pillar, and the Health information innovation and research, at Ministry of Health, for the invaluable feedback to the analyses and results presented in this paper. We appreciate the key informants in the districts and implementing partners for their time. The views expressed in this paper are solely of the authors and do not reflect that of the organizations they are affiliated with.

## Author Contributions

**Conceptualization:** Simon P. S. Kibira, Emily Evens, Rhoda K. Wanyenze, Fredrick E. Makumbi.

**Data curation:** Simon P. S. Kibira, Lilian Giibwa, Rogers Kagimu, Fredrick E. Makumbi.

**Formal analysis:** Simon P. S. Kibira, Emily Evens, Lilian Giibwa, Samantha Archie, Rawlance Ndejjo, Noel Namuhani, Martha Akulume, Sarah Nabukeera, Fredrick E. Makumbi.

**Funding acquisition:** Charles Olaro, Rhoda K. Wanyenze.

**Investigation:** Simon P. S. Kibira, Doreen Tuhebwe, Noel Namuhani, Martha Akulume, Fredrick E. Makumbi.

**Methodology:** Simon P. S. Kibira, Emily Evens, Lilian Giibwa, Doreen Tuhebwe, Rawlance Ndejjo, Sarah Nabukeera, Fredrick E. Makumbi.

**Software:** Simon P. S. Kibira, Lilian Giibwa, Fredrick E. Makumbi.

**Supervision:** Doreen Tuhebwe, Rawlance Ndejjo, Martha Akulume, Sarah Nabukeera.

**Validation:** Simon P. S. Kibira, Emily Evens, Andres Martinez, Rogers Kagimu, Charles Olaro, Samantha Archie.

**Visualization:** Lilian Giibwa, Fredrick E. Makumbi.

**Writing – original draft:** Simon P. S. Kibira, Emily Evens, Lilian Giibwa, Fredrick E. Makumbi.

**Writing – review & editing:** Simon P. S. Kibira, Emily Evens, Lilian Giibwa, Doreen Tuhebwe, Andres Martinez, Rogers Kagimu, Charles Olaro, Frederick Mubiru, Samantha Archie, Rawlance Ndejjo, Noel Namuhani, Sarah Nabukeera, Rhoda K. Wanyenze, Fredrick E. Makumbi.

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
