## [Decision Letter · Decision Letter 0]

13 Dec 2022

PGPH-D-22-01760

Uptake of Reproductive, Maternal and Child Health Services during the First Year of the COVID-19 Pandemic in Uganda

Dear Dr. Kibira,

Thank you for submitting your manuscript to PLOS Global Public Health. After careful consideration, we feel that it has merit but does not fully meet PLOS Global Public Health’s publication criteria as it currently stands. Therefore, we invite you to submit a revised version of the manuscript that addresses the points raised during the review process.

Please adress all the concern raised by the reviewers. 

We look forward to receiving your revised manuscript.

Kind regards,

Palash Chandra Banik, MPhil

Academic Editor

Journal Requirements:

2. Since your data is not available for proprietary reasons, please explain via email why the data is not available. Please also include the contact information for the third party organization that should be contacted should other researchers want to request access to this data and please include the full citation of where the data can be found. We also request that you verify with us via email that any researcher will be able to obtain the data set in the same manner that the you have obtained it. If you feel you are unwilling or unable to adhere to this policy, please explain your reasons by return email and your exemption request will be escalated to the editor for approval. Your exemption request will be handled independently and will not hold up the peer review process, but will need to be resolved should your manuscript be accepted for publication. One of the Editorial team will be in touch if they require more information.

Additional Editor Comments (if provided):

Reviewers' comments:

Reviewer's Responses to Questions

**Comments to the Author**

1. Does this manuscript meet PLOS Global Public Health’s publication criteria? Is the manuscript technically sound, and do the data support the conclusions? The manuscript must describe methodologically and ethically rigorous research with conclusions that are appropriately drawn based on the data presented.

Reviewer #1: Yes

Reviewer #2: Yes

Reviewer #3: Yes

2. Has the statistical analysis been performed appropriately and rigorously?

Reviewer #1: Yes

Reviewer #2: Yes

Reviewer #3: Yes

3. Have the authors made all data underlying the findings in their manuscript fully available (please refer to the Data Availability Statement at the start of the manuscript PDF file)?

Reviewer #1: Yes

Reviewer #2: Yes

Reviewer #3: Yes

4. Is the manuscript presented in an intelligible fashion and written in standard English?

Reviewer #1: Yes

Reviewer #2: Yes

Reviewer #3: Yes

5. Review Comments to the Author

Reviewer #1: I had a chance to review this study entitled, “Uptake of Reproductive, Maternal and Child Health Services during the First Year of the COVID-19 Pandemic in Uganda.”

This study describes that use of reproductive health (RH), maternal, newborn and child health (MNCH) services in Uganda, like many resource-limited countries, is suboptimal. Reasons for this are complex; however, service-delivery factors such as availability, quality, staffing, and supplies, among others, contribute substantially to low uptake. The COVID-19 pandemic threatened to exacerbate many of the existing challenges to delivery and use of high-quality RH and MNCH services. We conducted a mixed methods study, combining secondary analysis of routine health management information system (HMIS) data with exploratory key informant interviews (KII) to examine changes in health service uptake over the course of the pandemic and to understand adaptations in service delivery that were implemented in response to the pandemic. We analyzed HMIS data for four services (family planning, facility-based deliveries, antenatal visits, and immunization for children by one year), and compared them across four time periods: pre-COVID-19, partial lockdown, total lockdown and post lockdown.

I want to accept this study for publication. However, the authors need to revise it according to my suggestions. The title needs clarity with the design of the main study. I suggest authors revise their titles with better and suitable words. See the below-recommended studies to improve your TTLE and Abstract quality.

Abstract

First, I have some suggestions for the authors to enhance the quality of this innovative study. Please write a high-quality abstract, as it is the main door of the study. I suggest authors add Graphical Abstract in a meaningful way to reflect the whole idea. The abstract should be in a structured format.

Introduction section

This section needs improvement. Please read these studies, revise your abstract, and cite them in the introduction and literature part. Cite the suggested studies to improve the quality.

Su, Z., McDonnell, D., Wen, J., Kozak, M., Segalo, S., Li, X., Ahmad, J., Cheshmehzangi, A., Cai, Y., Yang, L., & Xiang, Y. T. (2021, Jan 5). Mental health consequences of COVID-19 media coverage: the need for effective crisis communication practices. Global Health, 17(1), 4. https://doi.org/10.1186/s12992-020-00654-4

Azadi, N. A., Ziapour, A., Lebni, J. Y., Irandoost, S. F., & Chaboksavar, F. (2021). The effect of education based on health belief model on promoting preventive behaviors of hypertensive disease in staff of the Iran University of Medical Sciences. Archives of Public Health, 79(1), 69. doi:10.1186/s13690-021-00594-4

Maqsood, A., Rehman, G., & Mubeen, R. (2021, 2021/11/01/). The paradigm shift for educational system continuance in the advent of COVID-19 pandemic: Mental health challenges and reflections. Current Research in Behavioral Sciences, 2, 100011. https://doi.org/https://doi.org/10.1016/j.crbeha.2020.100011

Abbas, J. (2020). The Impact of Coronavirus (SARS-CoV2) Epidemic on Individuals Mental Health: The Protective Measures of Pakistan in Managing and Sustaining Transmissible Disease. Psychiatr Danub, 32(3-4), 472-477. https://doi.org/10.24869/psyd.2020.472

Literature

In the emergence of the COVID-19 pandemic, healthcare systems have faced a tremendous pressure worldwide. This pandemic has affected all lifestyles. The study investigated an interesting research topic. The study offers interesting information and provides useful insight. I suggest authors cite latest literature related to COVID-19 pandemic effects to support the study. The suggested studies are;

Zeidabadi, S., Abbas, J., Mangolian Shahrbabaki, P., & Dehghan, M. (2022). The Effect of Foot Reflexology on the Quality of Sexual Life in Hemodialysis Patients: A Randomized Controlled Clinical Trial. Sexuality and Disability, 41(1), 1-12. doi:10.1007/s11195-022-09747-x

Rahmat, T. E., Raza, S., Zahid, H., Mohd Sobri, F., & Sidiki, S. (2022). Nexus between integrating technology readiness 2.0 index and students’ e-library services adoption amid the COVID-19 challenges: Implications based on the theory of planned behavior. J Educ Health Promot, 11(1), 50. doi:10.4103/jehp.jehp_508_21

NeJhaddadgar, N., Ziapour, A., Zakkipour, G., Abolfathi, M., & Shabani, M. (2020, Nov 13). Effectiveness of telephone-based screening and triage during COVID-19 outbreak in the promoted primary healthcare system: a case study in Ardabil province, Iran. Z Gesundh Wiss, 1-6. https://doi.org/10.1007/s10389-020-01407-8

Yoosefi Lebni, J., Moradi, F., Salahshoor, M. R., Chaboksavar, F., Irandoost, S. F., Nezhaddadgar, N., & Ziapour, A. (2020, Jul 2). How the COVID-19 pandemic effected economic, social, political, and cultural factors: A lesson from Iran. Int J Soc Psychiatry, 20764020939984. https://doi.org/10.1177/0020764020939984

Su, Z., McDonnell, D., Cheshmehzangi, A., Li, X., & Cai, Y. (2021). The promise and perils of Unit 731 data to advance COVID-19 research. BMJ Global Health, 6(4).

Methods and Results

Please read the suggested studies and improve your method design and result sections. Cite these studies in methods and results to support the literature.

Li, Zhenhuan , Dake Wang, Kaifeng Duan, and R. Mubeen. 2021. "Social media efficacy in crisis management: Effectiveness of non-pharmaceutical interventions to manage the COVID-19 challenges." Front Psychiatry 12 (1099):626134. doi: 10.3389/fpsyt.2021.626134.

Jiakui, C., Abbas, J., Najam, H., Liu, J., & Abbas, J. (2022). Green technological innovation, green finance, and financial development and their role in green total factor productivity: Empirical insights from China. Journal of Cleaner Production(381), 135131. doi:10.1016/j.jclepro.2022.135131

Abbas, J., Aman, J., Nurunnabi, M., & Bano, S. (2019). The Impact of Social Media on Learning Behavior for Sustainable Education: Evidence of Students from Selected Universities in Pakistan. Sustainability, 11(6), 1683. http://www.mdpi.com/2071-1050/11/6/1683

Aqeel, M., Rehna, T., Shuja, K. H. (2022). Comparison of Students' Mental Wellbeing, Anxiety, Depression, and Quality of Life During COVID-19's Full and Partial (Smart) Lockdowns: A Follow-Up Study at a 5-Month Interval. Front Psychiatry, 13, 835585. doi:10.3389/fpsyt.2022.835585

Farzadfar, F., Naghavi, M., Sepanlou, S. G., Saeedi Moghaddam, S., Dangel, W. J., Davis Weaver, N., . . . Larijani, B. (2022). Health system performance in Iran: a systematic analysis for the Global Burden of Disease Study 2019. The Lancet. doi:10.1016/s0140-6736(21)02751-3

Discussion

Discussion is explained well. Check grammar errors in this section.

Aqeel, M., Raza, S., & Aman, J. (2021). Portraying the multifaceted interplay between sexual harassment, job stress, social support and employees turnover intension amid COVID-19: A Multilevel Moderating Model. Foundation University Journal of Business & Economics, 6(2), 1-17. doi:https://fui.edu.pk/fjs/index.php/fujbe/article/view/551

Zhuang, D., Abbas, J., Al-Sulaiti, K., Fahlevi, M., Aljuaid, M., & Saniuk, S. (2022). Land-use and food security in energy transition: Role of food supply. Frontiers in Sustainable Food Systems, 6. doi:10.3389/fsufs.2022.1053031

Schmidt, C. A., Cromwell, E. A., Hill, E., Donkers, K. M., Schipp, M. F., Johnson, K. B., . . . Hay, S. I. (2022). The prevalence of onchocerciasis in Africa and Yemen, 2000-2018: a geospatial analysis. BMC Med, 20(1), 293. doi:10.1186/s12916-022-02486-y

Geng, J., Ul Haq, S., Ye, H., Shahbaz, P., Abbas, A., & Cai, Y. (2022). Survival in Pandemic Times: Managing Energy Efficiency, Food Diversity, and Sustainable Practices of Nutrient Intake amid COVID-19 Crisis. Frontiers in Environmental Science, 13, 945774. doi:10.3389/fenvs.2022.945774

Yu, S., Draghici, A., Negulescu, O. H., & Ain, N. U. (2022). Social Media Application as a New Paradigm for Business Communication: The Role of COVID-19 Knowledge, Social Distancing, and Preventive Attitudes. Frontiers in Psychology, 13. doi:10.3389/fpsyg.2022.903082

Implications

Explain this section effectively. It needs a better presentation related to the study topic.

Limitations

Discuss study’s limitations with a separate heading and discuss it briefly.

Policy recommendations

Policy recommendations are not sufficient at this stage of the manuscript. The authors must add a separate section for policy recommendations in the conclusion section. Also, add some exciting limitations regarding political factors for future studies.

Conclusion

The conclusion section needs improvement and authors need to expand it as it will improve the quality of this study. The English level needs some improvement to reach a satisfactory level, specifically the grammar. It should sufficiently meet quality to reach scientific merit for publication. I recommend that the authors describe the study's scientific contribution to the existing body of knowledge in the discussion section. How does this study’s implications provide useful information for the scientific readership? I endorse this manuscript for publication after minor corrections, as suggested.

Reviewer #2: The manuscript was done in excellent way. It addressed the issue under study very well. The comparison made during periods of pre-covid, partial lack down, total lock down and post lock down was clearly shown and stated. It was compared, analyzed and discussed satisfactorily. The figures and tables are self-explanatory and easy to grasp and understand. The statistical analysis done was great. The conclusion derived matched the exact finding of the result. The recommendations forwarded were in agreement with the result and they will have of great contribution. The study was ethically sound. Utilizing mixed methods of the quantitative and qualitative gives the paper better credit.

I am really excited to see such high quality, well-prepared and well-done large scale study.

There are only few minor points I need to forward.

1. Objective or the aim of the study should clearly be stated. Can you describe concisely what the research tries to achieve?

2. Would you mention the study period for the qualitative study (KII) that you did?

3. Try to write out all abbreviations in full as introduced in the text the first time, such as WHO (line 59 and 121), ICT (line 308) and IUD (line 339).

4. The methodology was discussed in all-inclusive way.

o However, it is unclear about how many health facilities were included in this study. Better also to explain the types of health facilities in which the secondary data were obtained; from hospitals, MCHs, clinics or health centers?

o Can you explain health facilities in Uganda providing those services (Reproductive, maternal and child health services) at national level? Or at least general information about how and where those services are provided, so that the readers can understand and have full image of the scope of the study.

o Were all health facilities in Uganda included in the study or they were just sampled? Would you briefly mention this? Let you be clear, specific and comprehensive. Can you very briefly describe about the administrative districts or states in Uganda too?

o You mentioned that you did pretest. Can you explain where the pretest was done and if any modifications made? (Line 178)

Reviewer #3: This mixed method research has provided a good sense about the situation of essential services affected by Covid 19. In the method section, qualitative data analysis should be well explained. Please illustrate how privacy during interview and anonymity of the research participants were maintained. Data management also needs to be clear.

The quantitative analysis of data is well narrated. But, qualitative data analysis require further elaborations especially difference between high performing and low performing districts and the challenges. The discussion section well captured challenges that should be described well in results section.

Conclusion looks good.

6. PLOS authors have the option to publish the peer review history of their article (what does this mean?). If published, this will include your full peer review and any attached files.

**Do you want your identity to be public for this peer review?** For information about this choice, including consent withdrawal, please see our Privacy Policy.

Reviewer #1: No

Reviewer #2: No

Reviewer #3: No

---

## [Decision Letter · Decision Letter 1]

3 Apr 2023

Uptake of Reproductive, Maternal and Child Health Services during the First Year of the COVID-19 Pandemic in Uganda: A Mixed Methods Study

PGPH-D-22-01760R1

Dear Dr. Kibira,

We are pleased to inform you that your manuscript 'Uptake of Reproductive, Maternal and Child Health Services during the First Year of the COVID-19 Pandemic in Uganda: A Mixed Methods Study' has been provisionally accepted for publication in PLOS Global Public Health.

Best regards,

Palash Chandra Banik, MPhil

Academic Editor

Reviewer Comments (if any, and for reference):

Reviewer's Responses to Questions

**Comments to the Author**

1. If the authors have adequately addressed your comments raised in a previous round of review and you feel that this manuscript is now acceptable for publication, you may indicate that here to bypass the “Comments to the Author” section, enter your conflict of interest statement in the “Confidential to Editor” section, and submit your "Accept" recommendation.

Reviewer #2: All comments have been addressed

2. Does this manuscript meet PLOS Global Public Health’s publication criteria? Is the manuscript technically sound, and do the data support the conclusions? The manuscript must describe methodologically and ethically rigorous research with conclusions that are appropriately drawn based on the data presented.

Reviewer #2: Yes

3. Has the statistical analysis been performed appropriately and rigorously?

Reviewer #2: Yes

4. Have the authors made all data underlying the findings in their manuscript fully available (please refer to the Data Availability Statement at the start of the manuscript PDF file)?

Reviewer #2: Yes

5. Is the manuscript presented in an intelligible fashion and written in standard English?

Reviewer #2: Yes

6. Review Comments to the Author

Reviewer #2: Great work. Well done. All comments were satisfactorily addressed.

7. PLOS authors have the option to publish the peer review history of their article (what does this mean?). If published, this will include your full peer review and any attached files.

**Do you want your identity to be public for this peer review?** For information about this choice, including consent withdrawal, please see our Privacy Policy.

Reviewer #2: No
